# Spatial Localization of a Transformer Robot Based on Ultrasonic Signal Wavelet Decomposition and PHAT-β-γ Generalized Cross Correlation

**DOI:** 10.3390/s24051440

**Published:** 2024-02-23

**Authors:** Hongxin Ji, Xinghua Liu, Jianwen Zhang, Liqing Liu

**Affiliations:** 1School of Electrical Engineering, China University of Mining and Technology, Xuzhou 221116, China; jihongx816@163.com (H.J.); jwzhang@cumt.edu.cn (J.Z.); 2College of Mechanical and Electronic Engineering, Shandong Agricultural University, Tai’an 271018, China; 3State Grid Tianjin Electric Power Research Institute, Tianjin 300180, China; liulq328@126.com

**Keywords:** robot localization, transformer, ultrasonic localization, wavelet decomposition, generalized cross correlation (GCC)

## Abstract

Because large oil-immersed transformers are enclosed by a metal shell, the on-site localization means it is difficult to achieve the accurate location of the patrol micro-robot inside a given transformer. To address this issue, a spatial ultrasonic localization method based on wavelet decomposition and PHAT-β-γ generalized cross correlation is proposed in this paper. The method is carried out with a five-element stereo ultrasonic array for the location of a transformer patrol robot. Firstly, the localization signal is decomposed into wavelet coefficients of different scales, which would realize the adaptive decomposition of the frequency of the localization signal from low frequencies to high frequencies. Then, the wavelet coefficients are denoised and reconstructed by using the semi-soft threshold function. Second, a modified phase transform-beta-gamma (PHAT-β-γ) method is used to calculate the exact time delay between different sensors by increasing the weights of the PHAT weighting function and introducing a correlation function. Finally, by using the proposed method, the accurate localization of the transformer patrol micro-robot is achieved with a five-element stereo ultrasonic array. The simulation and test results show that inside a transformer experimental oil tank (120 cm × 100 cm × 100 cm, L × W × H), the relative error of transformer patrol micro-robot spatial localization is within 4.1%, and the maximum localization error is less than 3 cm, which meets the requirement of engineering localization.

## 1. Introduction

The metal-enclosed nature of large transformers results in poor internal visibility. Without dismantling a large transformer, it is difficult to effectively visualize the internal insulation condition of a large transformer by only analyzing the dissolved gases in the oil (e.g., triple-ratio method, modified triple-ratio method, etc.). The finding of defects and faults inside oil-immersed transformers has been a difficult problem for substation maintenance due to the poor internal visibility caused by the metal sealing of oil-immersed transformers [1,2,3]. In order to accurately detect the internal defects of the transformer, the inspection is usually carried out by manually drilling into the transformer or lifting the cover [4,5]. But the method has three major problems: low efficiency, poor accuracy, and high risk.

With advances in artificial intelligence and micro-robotic technology, a micro-robot is developed for the patrol and inspection of transformer interiors, which allows for the more intuitive and faster analysis and localization of transformer faults. In 2018, ABB launched its first oil-immersed transformer internal patrol robot named Txplore [6,7]. Txplore is rectangular in shape and 18 × 20 × 24 cm in size, and it contains four propellers and multiple cameras. For the four propellers fixed on the shell of the robot, two in the horizontal direction drive the forward and steering of the robot, and the other two in the vertical direction control the upward and downward movement. Meanwhile, Shenyang Ligong University and Shenyang Institute of Automation of the Chinese Academy of Sciences have developed a similar spherical transformer robot named SSTIR with a diameter of 19 cm, with reference to the structure of underwater robots [8,9,10]. Referring to underwater vehicles, we have designed a new type of transformer patrol robot with fish fins [11,12]. The robot fish has a similar spherical structure, with two micro-propellers and a visual camera. Unlike the two robots above, our robot fish additionally has one swim bladder and four fins. The swim bladder is installed below the fish body and can accurately control the amount of oil absorbed or expelled, aiming to control the speed of uplift and downlift. For the four fins, each two are mounted on the upper and lower end of the fish body, respectively, which can control the deflection angle of the fish. With the help of the swim bladder and fins, the robot fish can glide even without power, effectively reducing power consumption and increasing operating hours.

When patrolling inside the transformer, the micro-robot is faced with a challenge: there are many equipment components inside the transformer, meaning the micro-robot passage is narrow. The micro-robot would easily dash against and damage internal equipment components. Therefore, how to accurately locate the spatial position of the micro-robot is the key to ensuring that the micro-robot smoothly avoids obstacles and accurately reaches the target point for defect detection. Since the transformer is completely sealed by a metal casing, the inside of the transformer is dark and filled with transformer oil, and it is difficult to use conventional methods such as LIDAR, vision cameras, and GPS to locate the micro-robot [13,14,15,16,17]. Considering that the ultrasonic localization method has the characteristics of no need for light, a high localization accuracy, and a wide detection range in transformer oil, it is an effective and feasible method for locating a micro-robot.

An important factor affecting the accuracy of ultrasonic localization is the time delay estimation of an ultrasonic signal. The commonly used methods for time delay estimation include: the rising-edge triggering method, the threshold triggering method, basic cross correlation, generalized cross correlation, adaptive filtering, a neural network, deep learning, and so on [18,19,20,21,22]. Among them, the rising-edge trigger method and threshold trigger method are the least computationally intensive methods for time delay estimation, which are generally applicable to the occasions where the waveform profile is clear and has no distortion or small distortion. The basic cross correlation method achieves time delay estimation by peak detection of the correlation function, which has the advantages of fast operation, good robustness, and a certain degree of waveform distortion tolerance. It is the most widely applicable method for time delay estimation at present. However, with the increase in noise components and the deterioration of waveform quality in modern detection fields, the adaptation range of the basic cross correlation method is limited. In order to improve the performance of time delay estimation, Knapp and Carter et al. [23] proposed a generalized cross correlation algorithm, which can serve to sharpen the cross correlation function curve and improve the performance of time delay estimation by pre-whitening the signal. The commonly used generalized cross correlation algorithms include Roth, PHAT, SCOT, HB, and maximum likelihood estimation [12].

Although generalized cross correlation algorithms can improve the performance of delay estimation to a certain extent, these methods are often designed for specific application scenarios and signal characteristics. The computational parameters used in these methods are often fixed and cannot be adjusted according to the signal changes. Adaptive filtering, neural networks, and deep learning methods can theoretically obtain more accurate performances in time delay estimation by iterative computation and learning the signal, but they suffer from the problem of large computation and are currently difficult to be applied in the field of real-time localization. Considering the complex variation of the ultrasonic localization signal and the demand for the three-dimensional real-time localization of a micro-robot, the above methods are not yet able to solve the spatial localization problem of a patrol micro-robot inside a power transformer.

Therefore, for the problems of strong electromagnetic interference and complex noise inside power transformers, the paper proposes a delay estimation algorithm based on ultrasonic signal wavelet decomposition and PHAT-β-γ generalized cross correlation, which would achieve the spatial localization of a transformer patrol micro-robot using a five-element stereo ultrasonic array.

## 2. Micro-Robot Localization Method Based on the Optimization of Ultrasonic Time Delay Estimation

### 2.1. Ultrasonic Signal Decomposition and Reconstruction Based on Wavelet Transform

Wavelet transform (WT) is a powerful signal processing tool developed in recent years, which can represent the local characteristics of signals in both time and frequency domains and is a suitable method for dealing with signals with transient and abrupt characteristics.

The wavelet transform of any signal *x*(*t*) is defined as the inner product of the signal and the wavelet basis function, i.e.,
(1)$$W_{x}(a,\tau )=\int_{-\infty }^{+\infty }x(t)\psi_{a\tau }^{*}(t)dt$$
(2)$$\psi_{a\tau }(t)=\frac{1}{\sqrt{a}}\psi (\frac{t-\tau }{a})a>0$$
where $W_{x}(a,\tau )$ is the wavelet coefficient; $\psi_{a\tau }(t)$ is the wavelet basis function; a is the scale factor; τ is the translation parameter, whose value can be positive or negative; and * denotes the complex conjugate.

The wavelet transform obtains the frequency characteristics of an ultrasound signal by scaling the width of the wavelet basis function and the time information of the signal by translating the wavelet basis function. The wavelet coefficients obtained by wavelet transform represent the correlation degree between wavelets and ultrasound signals, and the calculation process of wavelet coefficients is shown in Figure 1.

The wavelet transform only decomposes the low-frequency components of the signal, so it can characterize the signal well when most of the key components in the signal are low-frequency components but cannot well characterize and decompose the signal with a large number of high-frequency components. As an extension of wavelet transform, wavelet packet transform provides a more flexible decomposition method for addressing this problem. Since it can decompose both low-frequency components and high-frequency components, wavelet packet transform can achieve better time-frequency localization analysis for signals with a large number of medium- and high-frequency components.

The fast algorithm for wavelet packets is divided into two parts: decomposition and reconstruction. The schematic diagram of the three-layer decomposition of wavelet packets is shown in Figure 2.

The three-layer wavelet packet decomposition of signal s is to decompose both the low-frequency and high-frequency components of the signal to the next layer and then continue the decomposition, which improves the time-frequency resolution of signal processing. The ordering of the individual frequency bands after decomposition is an important issue in wavelet packet decomposition. Take the three-layer wavelet packet decomposition as an example; the decomposition law is: if (m, n) denotes the mth layer and the nth decomposition node, the low-frequency and high-frequency parts generated by each decomposition are denoted by a and d, respectively. If the last bit of the node is a, the decomposed low-frequency part is on the left side and the high-frequency part is on the right side; if the last bit of the node is d, the opposite is true.

The essence of wavelet packet threshold denoising is the process of suppressing the useless part of the signal and enhancing the useful part. The wavelet packet threshold denoising process is: (a) the decomposition process, i.e., a wavelet is selected to perform n-layer wavelet decomposition on the signal; (b) the thresholding process, i.e., the coefficients of each layer of the decomposition are denoised according to the threshold values; (c) the reconstruction process, i.e., the denoised wavelet coefficients are wavelet-reconstructed to obtain the denoised signal. In the process of wavelet packet thresholding denoising, it is crucial to choose the appropriate wavelet basis and threshold function. Currently, the widely used threshold functions are the hard threshold function and soft threshold function.

The hard threshold function is expressed as
(3)$$W_{new}(j,k)=\left\{\begin{array}{l} W_{x}(j,k),\;\;\left|W_{x}(j,k)\right|\geq \lambda \\ 0,\;\;\;\;\left|W_{x}(j,k)\right|\leq \lambda \end{array}\right.$$
where the wavelet coefficient remains constant when the absolute value of the wavelet coefficient is greater than a given threshold and zero when it is less than the threshold. Hard thresholding functions can preserve the local features of the signal relatively better.

The soft threshold function is expressed as
(4)$$W_{new}(j,k)=\left\{\begin{array}{l} \mathrm{sgn}(W_{x}(j,k))(\left|W_{x}(j,k)\right|-\lambda ),\;\left|W_{x}(j,k)\right|\geq \lambda \\ 0\;\;\;\left|W_{x}(j,k)\right|\leq \lambda \end{array}\right.$$
where sgn() is the sign function. When the absolute value of the wavelet coefficient is greater than the given threshold, the wavelet coefficient is subtracted from the threshold; when it is less than the threshold, the wavelet coefficient is equal to zero. The signal processed by the soft threshold function will be smoother.

The hard threshold function is discontinuous at −λ and λ, causing the reconstructed signal to oscillate around −λ and λ, which does not have the smoothness of the original signal. The wavelet coefficients obtained by the soft threshold function have better continuity but will compress the signal and affect the approximation of the reconstructed signal to the real signal. To this end, a semi-soft threshold function is constructed by combining the advantages and disadvantages of soft and hard threshold functions.
(5)$$W_{new}(j,k)=\left\{\begin{array}{l} W_{x}(j,k),\;\left|W_{x}(j,k)\right|\geq \lambda_{2}\\ \mathrm{sgn}\left[W_{x}(j,k)\right]\lambda_{2}{\left[\frac{\left|W_{x}(j,k)\right|-\lambda_{1}}{\lambda_{2}-\lambda_{1}}\right]}^{h},\;\;\lambda_{1}<\left|W_{x}(j,k)\right|<\lambda_{2}\\ 0,\;\;\;\left|W_{x}(j,k)\right|\leq \lambda_{1} \end{array}\right.$$
where sgn() is the sign function, $\lambda_{1}$ is the lower threshold of the threshold function, and $\lambda_{2}$ is the upper threshold of the threshold function and is set to $\lambda_{2}=2\lambda_{1}$.

The semi-soft thresholding function zeroes out the wavelet coefficients smaller than the lower threshold and keeps the wavelet coefficients larger than the upper threshold. The wavelet coefficient function transform between the upper and lower thresholds solves the continuity problem of hard thresholding and the compression problem of soft thresholding.

In addition to the selection of a suitable threshold function, wavelet packet thresholding noise reduction also requires the determination of a threshold value. The threshold *thr* used in this paper is calculated as follows.
(6)$$thr=\sigma \sqrt{2\log(N)}$$
where σ is the standard deviation of the noise in the signal (mean squared deviation), and N is the length of the signal, $\lambda_{1}=0.75thr$.

### 2.2. Optimization of the Ultrasonic Signal Time Delay Estimation Algorithm for the Micro-Robot

Assuming that the ultrasound signal emitted by the micro-robot is s(t), the ultrasound signal is affected by attenuation, time delay, noise, and other factors during propagation. Therefore, the signals received by the ultrasound receiving sensor 1 and receiving sensor 2 can be expressed as
(7)$$\left\{\begin{array}{l} x_{1}(t)=a_{1}s(t-\tau_{1})+n_{1}(t)\\ x_{2}(t)=a_{2}s(t-\tau_{2})+n_{2}(t) \end{array}\right.$$
where $a_{1}$ and $a_{2}$ are the attenuation coefficients of the acoustic waves, $n_{1}(t)$ and $n_{2}(t)$ are the white noise, and $\tau_{1}$ and $\tau_{2}$ denote the time needed for the acoustic waves to propagate to the ultrasound receiving transducers 1 and 2, respectively. Then, $\tau_{12}=\tau_{1}-\tau_{2}$ is the time difference between the ultrasound signals arriving at the two transducers.

Based on the theory of the propagation of ultrasonic signals in transformer oil, the cross correlation time delay estimation model for multi-channel ultrasonic sensors is shown below.
(8)$$R_{x_{1}x_{2}}(\tau )=E[x_{1}(t)x_{2}(t+\tau )]$$

Substituting Equation (7) into Equation (8) yields
(9)$$R_{x_{1}x_{2}}(\tau )=a_{1}a_{2}R_{ss}(\tau -\tau_{12})+a_{1}R_{sn_{2}}(\tau -\tau_{1})+a_{2}R_{sn_{1}}(-\tau -\tau_{2})+R_{n_{1}n_{2}}(\tau )$$
where $R_{ss}(\tau -\tau_{12})$ is the correlation function between the effective signals (i.e., the signal of attenuation and delay), $R_{sn_{2}}(\tau -\tau_{1})$ and $R_{sn_{1}}(-\tau -\tau_{2})$ is the correlation function between the effective signal and the noise, and $R_{n_{1}n_{2}}(\tau )$ is the correlation function between the noises. In general, the signal and the noise and the noise and the noise are all weakly correlated or uncorrelated, i.e.,
(10)$$R_{sn_{2}}(\tau -\tau_{1})=R_{sn_{1}}(-\tau -\tau_{2})=R_{n_{1}n_{2}}(\tau )\approx 0$$ Therefore, the correlation function can be abbreviated as
(11)$$R_{x_{1}x_{2}}(\tau )=a_{1}a_{2}R_{ss}(\tau -\tau_{12})$$

According to the properties of the self-correlated function $R_{ss}(\tau -\tau_{12})$ in Equation (11), the maximum value of $R_{x_{1}x_{2}}(\tau )$ is taken when $\tau =\tau_{12}$.

According to the Wiener–Sinchin theorem, the cross correlation function and its cross-power spectral density are Fourier transform pairs. Then, the correlation functions of $x_{1}(t)$ and $x_{2}(t)$ can be further expressed as
(12)$$R_{x_{1}x_{2}}(\tau )=\frac{1}{2\pi }\int_{-\infty }^{+\infty }G_{x_{1}x_{2}}(\omega )e^{j\omega t}d\omega$$
where $G_{x_{1}x_{2}}(\omega )$ is the cross-power density spectrum between the signals $x_{1}(t)$ and $x_{2}(t)$ acquired by the ultrasound transducer, and $R_{x_{1}x_{2}}(\tau )$ is the horizontal coordinate at the corresponding maximum peak, which is also the required time delay estimate.

In the actual time-delay estimation process, the noises in the signal are often not completely independent of each other due to the limited length of the signal collected by the ultrasonic receiver transducer, and the signal will also contain the ultrasonic echo component. In the case of a relatively low signal-to-noise ratio, it will cause the peaks of the correlation function to become obscure or even pseudo-peak, which greatly reduces the accuracy of the time delay estimation.

To address the above problem, a common method is to weight the power spectrum in Equation (12) and use the frequency domain weighting function to filter the signal, sharpen the correlation, and improve the accuracy of the time delay estimation. This results in the generalized intercorrelation function.
(13)$$R_{x_{1}x_{2}}(\tau )=\frac{1}{2\pi }\int_{-\infty }^{+\infty }X_{1}(\omega ){X_{2}}^{*}(\omega )\Psi (\omega )e^{j\omega t}d\omega$$
where Ψ(ω) is the weighting function. The weighting function of PHAT GCC is
(14)$$\Psi_{PHAT}(\omega )=\frac{1}{\left|Gx_{1}x_{2}(\omega )\right|}$$
where *Gx*_1_*x*_2_(*ω*) is the cross-power spectral density of signal *x*_1_ and signal *x*_2_.

At low signal-to-noise ratios, the PHAT-based weighting function has better performances under the same conditions. However, as the signal-to-noise ratio decreases further, its time delay estimation accuracy also decreases. For this reason, the PHAT weighting function is improved in this paper by increasing the weights, and its expression is
(15)$$\Psi_{PHAT-\beta }(\omega )=\frac{1}{{\left|Gx_{1}x_{2}(\omega )\right|}^{\beta }}$$
where *β* is an exponential adjustment factor in the range of [0,1].

When the signal energy is small, the denominator term of the weighting function tends to 0, while the overall weighting function tends to infinity, which will result in a large error. For this reason, a non-zero factor needs to be added to the denominator of the weighting function, and the mode-squared coherence function ${\left|\gamma \right|}^{2}$ is chosen as the non-zero factor in this paper.
(16)$$\Psi_{PHAT-\beta -\gamma }(\omega )=\frac{1}{{\left|Gx_{1}x_{2}(\omega )\right|}^{\beta }+{\left|\gamma \right|}^{2}}$$
where $\gamma =\frac{G_{x_{1}x_{2}}(\omega )}{\sqrt{G_{x_{1}x_{1}}(\omega )G_{x_{2}x_{2}}(\omega )}}$. In the actual system, the signals received by the two sensor arrays are neither perfectly coherent nor perfectly incoherent, so $0<\gamma^{2}(\omega )<1$.

### 2.3. Localization Method of a Micro-Robot in a Power Tranformer

The ultrasonic sensor array consists of five omnidirectional waterproof piezoelectric ceramic ultrasonic probes, four of which are arranged in a quadrilateral on a plane, and another ultrasonic probe is directly above the intersection of the diagonals of the quadrilateral, forming a quadrilateral cone, as shown in Figure 3. The five ultrasonic probe coordinates are U_1_ (0, 0, h), U_2_ (b, b, 0), U_3_ (−b, b, 0), U_4_ (−b, −b, 0), and U_5_ (b, −b, 0), respectively.

Assuming that the coordinate position of the ultrasonic source on the body of the micro-robot is *T*_0_(*x_f_*, *y_f_*, *z_f_*), based on the principle that the sound waves propagate in the material along the path of minimum energy attenuation, and based on the five-element ultrasonic array configuration in Figure 3, the distance from the micro-robot to the ultrasonic probe can be obtained by the product of the sound velocity and the time delay, as shown in Equation (17).
(17)$$\left\{\begin{array}{c} \sqrt{{\left(x_{f}-b\right)}^{2}+{\left(y_{f}-b\right)}^{2}+z_{f}^{2}}-\sqrt{x_{f}^{2}+y_{f}^{2}+{\left(z_{f}-h\right)}^{2}}=V_{c}(t_{2}-t_{1})\\ \sqrt{{\left(x_{f}+b\right)}^{2}+{\left(x_{f}-b\right)}^{2}+z_{f}^{2}}-\sqrt{x_{f}^{2}+y_{f}^{2}+{\left(z_{f}-h\right)}^{2}}=V_{c}(t_{3}-t_{1})\\ \sqrt{{\left(x_{f}+b\right)}^{2}+{\left(x_{f}+b\right)}^{2}+z_{f}^{2}}-\sqrt{x_{f}^{2}+y_{f}^{2}+{\left(z_{f}-h\right)}^{2}}=V_{c}(t_{4}-t_{1})\\ \sqrt{{\left(x_{f}-b\right)}^{2}+{\left(y_{f}+b\right)}^{2}+z_{f}^{2}}-\sqrt{x_{f}^{2}+y_{f}^{2}+{\left(z_{f}-h\right)}^{2}}=V_{c}(t_{5}-t_{1}) \end{array}\right.$$
where *t*_i_ is the moment when the sensor *U*_i_ receives the signal from the sound source, and *V_c_* is the speed of the sound wave in the oil.

The analytical solution of the three-dimensional coordinate position of the sound source of the patrolling micro-robot, calculated by Equation (17), is shown as follows.
(18)$$\left\{\begin{array}{l} x_{f}=\frac{d_{31}^{2}-d_{21}^{2}+2r_{0}(d_{31}-d_{21})}{4b}\\ y_{f}=\frac{d_{41}^{2}-d_{31}^{2}+2r_{0}(d_{41}-d_{31})}{4b}\\ z_{f}=h+\sqrt{r_{0}^{2}-x_{f}^{2}-y_{f}^{2}} \end{array}\right.$$ The parameters in the formula are as follows:(19)$$r_{0}=\frac{d_{41}^{2}-d_{51}^{2}+d_{21}^{2}-d_{31}^{2}}{2(d_{31}-d_{21}+d_{51}-d_{41})}$$
(20)$$d_{21}=V_{c}(t_{2}-t_{1}$$
(21)$$d_{31}=V_{c}(t_{3}-t_{1})$$
(22)$$d_{41}=V_{c}(t_{4}-t_{1})$$
(23)$$d_{51}=V_{c}(t_{5}-t_{1})$$

## 3. Simulation Validation of the 3D Spatial Localization Method for Transformer Micro-Robots

### 3.1. Ultrasonic Signal Characteristics

Simulation work was carried out to verify the performance of the method. The emitted ultrasonic signal consists of three sinusoidal signals with a frequency of 180 kHz, and the noise signal is Gaussian white noise. The sampling frequency is 2.5 MSa/s, and the number of sampling points is 4000. To simulate the effect of noise on the signal, Gaussian white noise with SNR = −10 dB was added to the signal, as shown in Figure 4.

The spectrum of the ultrasonic signal is shown in Figure 5. The frequency range of this signal is 160–200 kHz, there are differences in the propagation path of the sound source to the three transducers, and the degree of attenuation is not consistent during the propagation. There is a difference in the frequency band range of the ultrasonic signal received by each sensor.

The ultrasonic signal with noise is decomposed into three wavelet packets, and the decomposed wavelet coefficients are shown in Figure 6.

Figure a3i indicates the third layer, the i-th wavelet packet decomposition node, and it can be seen from the figure that the wavelet coefficients at the a30 and a31 decomposition nodes contain a larger useful signal component, and the other decomposition nodes contain a smaller useful signal component. For each wavelet coefficient, the threshold is first estimated using the universal thresholding (sqtwolog principle) method, and then each wavelet coefficient is denoised using the hard thresholding function, soft thresholding function, and semi-soft thresholding function, respectively; then, the denoised wavelet coefficients are reconstructed. The reconstructed signal is shown in Figure 7.

In order to effectively evaluate the noise reduction ability of the wavelet denoising algorithm with the semi-soft threshold function and the traditional wavelet threshold denoising algorithm for signals in this paper, the signal-to-noise ratio (SNR), root mean square error (RMSE), and waveform similarity coefficient (NCC) are used as evaluation indices to determine the superiority of the noise reduction effect of ultrasonic signals, where the SNR is defined as
(24)$$SNR=10\mathrm{lg}\frac{\sum_{i=1}^{n}x^{2}(i)}{\sum_{i=1}^{n}{\left(x(i)-\overset{-}{x}(i)\right)}^{2}}$$

The RMSE is defined as
(25)$$RMSE=\sqrt{\frac{1}{n}\sum_{i=1}^{n}{\left(x(i)-\overset{-}{x}(i)\right)}^{2}}$$

The NCC is defined as
(26)$$NCC=\frac{\sum_{i=1}^{n}x(i)\cdot \overset{-}{x}(i)}{\sqrt{(\sum_{i=1}^{n}x^{2}(i))\cdot (\sum_{i=1}^{n}\overset{-}{x^{2}}(i))}}$$
where x(i) denotes the original signal and $\overset{-}{x}(i)$ denotes the denoised signal.

The calculated results of the signal noise reduction indexes SNR, RMSE, and NCC for different denoising methods are shown in Table 1.

The larger the value of SNR, the smaller the proportion of noise in the signal and the better the denoising effect; the smaller the RMSE, the smaller the distortion of the signal; and the closer the NCC value is to 1, the better the denoising effect of the simulated signal. As can be seen in Table 1, compared with the traditional threshold functions, the semi-soft threshold function has the largest value of SNR, the smallest RMSE, and the closest NCC value to 1. It shows that this semi-soft threshold function has a better denoising effect on signal denoising.

### 3.2. Time Delay Estimation

The arrival time of the sine signal in signal 1 is the 300th sample point, and the arrival time of the sine signal in signal 2 is the 500th sample point. Therefore, the delay time of the two signals is 200 samples. Figure 8 and Figure 9 show the TDE results for the basic intercorrelation, the PHAT-β generalized intercorrelation, and the PHAT-β-γ generalized intercorrelation, respectively.

A comprehensive comparison of Figure 8 shows that the TDE performance gradually becomes better as the exponential regulator β increases in the range of [0, 0.8]. The best results are obtained when the exponential regulator β is 0.6 and 0.8. When the exponential regulator β is larger than 0.8, the TDE results become worse. Therefore, it is necessary to select the optimal exponential regulator according to the signal characteristics and noise conditions in practical applications. According to the signal and noise characteristics set in the simulation, the optimal value of the exponential regulator β is 0.8. Meanwhile, it can be seen from Figure 9 that when the signal energy is small, the TDE performance can be effectively improved by adding a non-zero factor to the denominator of the weighting function.

## 4. Experimental Testing and Verification of the 3D Spatial Localization Effect of the Patrol Micro-Robot

### 4.1. Test Platform for Patrol Micro-Robot 3D Spatial Localization

In order to verify the practicability of the three-dimensional positioning method proposed in this paper, an experimental test platform is built in this paper. The test platform mainly includes five parts: the transformer inspection robot, ultrasonic positioning array, and data acquisition and processing platform. The test platform is shown in Figure 10.

The overall structure of the patrol micro-robot is shown in Figure 11. The patrol micro-robot mainly includes the: #1—robot main body shell, #2—ultrasonic emission sensor, #3—vision device, #4—infrared range module, #5—robot vertical propeller propulsion device, #6—robot horizontal propeller propulsion device, #7—pressure sensor, and robot control system, etc. The main body of the patrol micro-robot is a spherical sealed structure, and an ultrasonic emission sensor is installed on the top of the main body of the patrol micro-robot, which emits ultrasonic signals at regular intervals, is received by the ultrasonic array installed at the tank, and is mainly used for the three-dimensional localization of the patrol micro-robot. There are two infrared distance measurement modules installed in the front of the main body of the patrol micro-robot, which can be used to measure the distance of surrounding obstacles; there are four vertical propeller propulsion devices installed around the middle of the main body of the patrol micro-robot, through which the vertical propeller propulsion devices can control the floating; and there are two horizontal propeller propulsion devices installed in the middle and lower part of the main body of the patrol micro-robot, through which the horizontal propeller propulsion devices can control the forward and turning of the patrol micro-robot; there is a depth pressure sensor installed in the bottom of the main body of the patrol micro-robot, which can be used to measure the diving depth. There is the robot control system inside the main body of the patrol micro-robot, which mainly includes the motor drive and control module, sensor acquisition module, position detection module, wireless transmission module, power supply module, etc.

The robot is spherical, with a diameter (TR1) of 123 mm. The total height (TH) of the transformer patrol robot is 130 mm, the total width (TW) of the robot is 127 mm, and the maximum diameter (TR2) is 139 mm.

There is an ultrasonic emission sensor installed on the top of the main body of the patrol micro-robot, which emits ultrasonic signals at regular intervals and is received by the ultrasonic array, which consists of five omnidirectional waterproof piezoelectric ceramic ultrasonic probes, four of which are arranged in a quadrilateral on a plane, and the other ultrasonic probe is directly above the intersection of the diagonals of the quadrilateral, forming a quadrilateral cone, which has a nominal. The ultrasonic localization transducer array is shown in Figure 12. The ultrasonic probe has a diameter of 12.7 mm, an operating frequency of 160~200 kHz, a resonance frequency of 180 kHz, a transducer vertical directivity of −90~90°, and a transducer horizontal directivity of 0~360°. The coordinates of the five ultrasonic probes are U1 (0 mm, 0 mm, 110 mm), U2 (110 mm, 110 mm, 0 mm), U3 (−110 mm, 110 mm, 0 mm), U4 (−110 mm, −110 mm, 0 mm), and U5 (110 mm, −110 mm, 0 mm), respectively.

### 4.2. Experiment and Data Acquisition Process

In the experiment, the patrol micro-robot was moved to position 1 (−220 mm, 0 mm, 690 mm), at which time the ultrasonic transducer on top of the patrol micro-robot emitted a set of ultrasonic signals with three consecutive sinusoidal signals at a frequency of 180 kHz every 500 ms. At the same time, the ultrasonic signals were received by the ultrasonic array, and the data were synchronously acquired by the data acquisition device. The parameters of the data acquisition device are set as follows: the sampling rate of each channel is 2.5 MSa/s, the acquisition depth is 4096 points, and the triggering method is rising edge triggering. The ultrasonic signals collected by the acquisition device are shown in Figure 13.

The five time-domain waveforms in Figure 13 are the ultrasonic signals collected by sensor U1, sensor U2, sensor U3, sensor U4, and sensor U5, respectively. The ultrasonic signal contains system noise, which mainly includes: mechanical equipment vibration, electromagnetic noise, and environmental white noise. The middle part of the received waveform is the ultrasonic signal, which can be seen from the contour as three sine signals emitted, but the amplitude of the sine signal changes during propagation, and the signal contains large noise, making it difficult to obtain the first wave position of the signal accurately. The ultrasonic-shaped signals obtained from the five sensors were decomposed into three wavelet packets. Taking the ultrasonic-shaped signal from sensor U5 as an example, the wavelet coefficients after the decomposition are shown in Figure 14.

The wavelet coefficients at the a30 and a31 decomposition nodes in the Figure 14 contain a larger useful signal component, and the other decomposition nodes contain a smaller useful signal component. The wavelet coefficients are first denoised using the general thresholding method; then, each wavelet coefficient is denoised using the semi-soft thresholding function; and finally, the denoised wavelet coefficients are re-constructed, and the reconstructed ultrasonic waveform signal is shown in Figure 15.

After the wavelet denoising, the signal-to-noise ratio of the ultrasonic signal is significantly improved, which is beneficial to the calculation of time delay (TDE). The ultrasonic signals of sensor No. 2, sensor No. 3, sensor No. 4, and sensor No. 5 were denoised and reconstructed using the basic cross correlation and generalized cross correlation algorithms, respectively, and the TDE was calculated and compared with the signal of sensor No. 1, respectively. The obtained TDE results are shown in the following figures.

As can be seen in Figure 16, there are multiple peaks in the curves obtained using the basic cross correlation (BCC) algorithm, and the magnitudes of the maximum peaks are close to each other, as shown in the left panels of Figure 16a,b,d. Such peaks of similar magnitudes will increase the error of the time delay estimation to some extent. Compared with the results of the basic correlation, the GCC used in the paper has the effect of sharpening the peaks, and the peaks of the curves obtained are more obvious; thus, the delay estimates can be obtained more accurately.

From the results obtained using the generalized cross correlation algorithm in Figure 16a–d, it can be determined that the estimated time delay between sensor 2 and sensor 1 is Δt21 = 58.4 μs, the estimated time delay between sensor 3 and sensor 1 is Δt31 = 114 μs, the estimated time delay between sensor 4 and sensor 1 is Δt41 = 114 μs, and the estimated time delay between sensor 5 and sensor 1 is Δt51 = 114 μs.

To verify the effectiveness of the proposed wavelet decomposition and PHAT-β-γ generalized cross correlation algorithm based on the ultrasonic signal when the sound source of the patrol micro-robot is at different positions, the micro-robot is moved from the original position 1 (−220 mm, 0 mm, 690 mm) to position 2 (0 mm, 0 mm, 690 mm) and position 3 (220 mm, 0 mm, 690 mm). The ultrasonic signals were repeatedly acquired 10 times at different positions, and the time delay estimates were obtained by wavelet packet denoising and the generalized correlation algorithm. Finally, the spatial position of the micro-robot was estimated based on the five-element ultrasonic array localization method.

From the repeated localization results of the patrol micro-robot at position 1 in Table 2, we can see that the maximum X-axis localization error of the 3D spatial localization method of the patrol micro-robot at sound source position 1 is 12 mm, and the average X-axis localization error is 5 mm; the maximum Y-axis localization error at sound source position 1 is 7 mm, and the average Y-axis localization error is 1.8 mm; the maximum Z-axis localization error at sound source position 1 is 27 mm, and the average Z-axis localization error is 14.7 mm; the maximum relative localization error of the micro-robot patrolling inside the transformer at location 1 is 4.1%, and the average relative localization error is 2.28%.

From the repeated localization results of the patrol micro-robot at position 2 in Table 3, we can see that the maximum X-axis localization error of the 3D spatial localization method of the patrol micro-robot at sound source position 2 is 14 mm, and the average X-axis localization error is 11 mm; the maximum Y-axis localization error at sound source position 2 is 13 mm, and the average Y-axis localization error is 7.3 mm; the maximum Z-axis localization error at sound source position 2 is 23 mm, and the average Z-axis localization error is 10.9 mm; the maximum relative localization error of the micro-robotic patrolling inside the transformer at location 2 is 3.53%, and the average relative localization error is 2.72%.

From the repeated localization results of the patrol micro-robot at position 3 in Table 4, we can see that the maximum X-axis localization error of the 3D spatial localization method of the patrol micro-robot at sound source position 3 is 9 mm, and the average X-axis localization error is 3.5 mm; the maximum Y-axis localization error at sound source position 3 is 3 mm, and the average Y-axis localization error is 1.1 mm; the maximum Z-axis localization error at sound source position 3 is 28 mm, and the average Z-axis localization error is 16 mm; the maximum relative localization error of the micro-robot patrolling inside the transformer at location 3 is 3.94%, and the average relative localization error is 2.38%.

The above experimental results verify the effectiveness of the proposed micro-robot spatial localization method based on ultrasonic signal wavelet decomposition and PHAT-β-γ generalized cross correlation and a five-element stereo ultrasonic array at different locations. The localization results show that the 3D spatial relative localization error of the micro-robot is within 4.1%, and the maximum localization error is less than 3 cm, which meets the engineering application of the current patrol micro-robot.

## 5. Conclusions

In this paper, a new method for the spatial localization of a transformer micro-robot based on ultrasonic signal wavelet decomposition, a PHAT-β-γ generalized cross correlation algorithm, and five-element stereo ultrasonic array localization is proposed. In order to verify the performance of the method, simulation work is carried out. The simulation results show that: the ultrasonic localization simulation signal is significantly improved by using the wavelet packet decomposition and semi-soft threshold function denoising reconstruction proposed in this paper, and the signal-to-noise ratio of the ultrasonic signal is significantly improved; the TDE performance gradually becomes better as the exponential regulator β increases in the PHAT-β-γ generalized cross correlation algorithm; when the exponential regulator β is 0.6 and 0.8, the best results are obtained, and the time delay estimation results become gradually worse after the exponential regulator β exceeds 0.8; when the signal energy is small, the TDE performance can be effectively improved by adding a non-zero factor to the denominator of the weighting function.

Based on the simulation verification, this paper builds an experimental test platform for validating the three-dimensional spatial localization effect of micro-robot patrolling inside the transformer and verifies the effectiveness of the spatial localization method proposed in this paper when the sound source of the patrol micro-robot is at different locations through experiments. The localization results show that the relative three-dimensional spatial localization error of the micro-robot is within 4.1%, and the maximum localization error is less than 3 cm, which meets the current engineering application of the inspection robot.

## Figures and Tables

**Figure 1 sensors-24-01440-f001:**
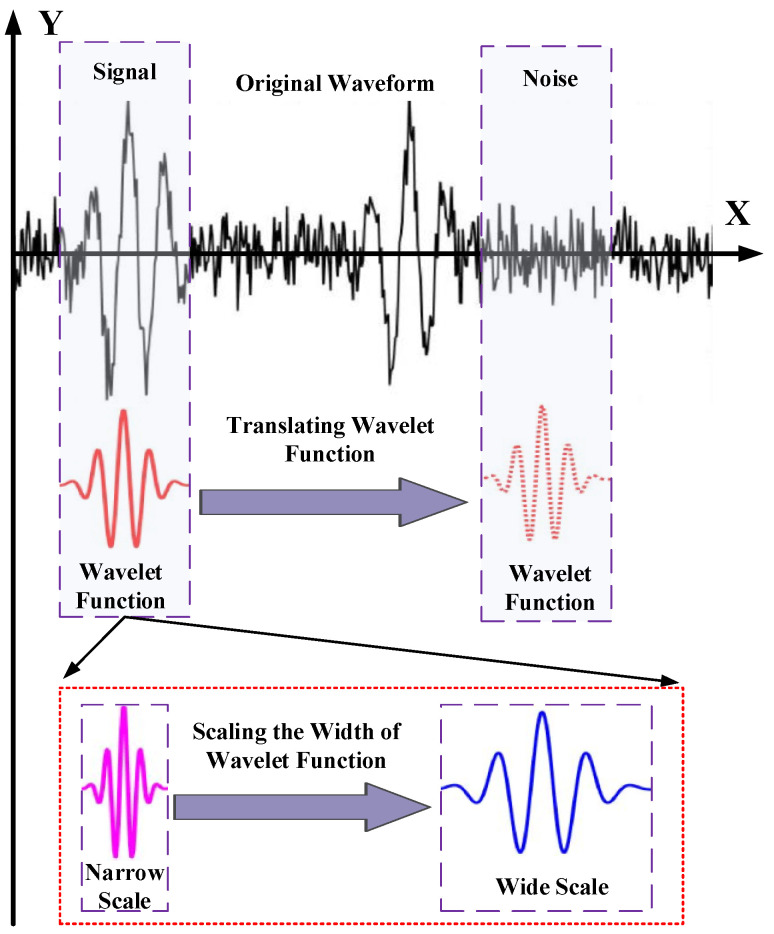
Wavelet coefficient calculation process.

**Figure 2 sensors-24-01440-f002:**
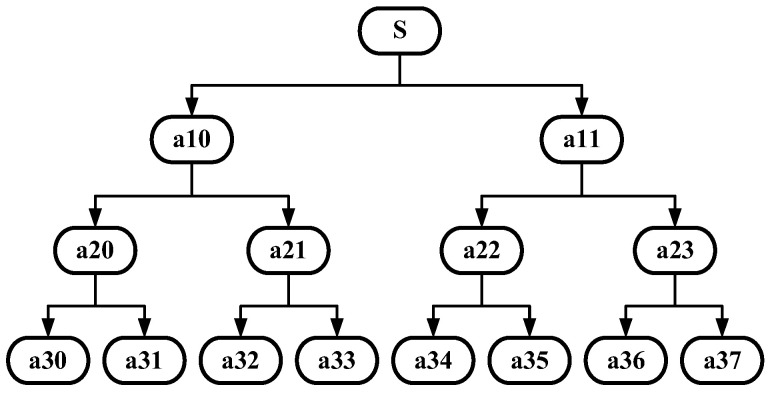
Three-layer wavelet packet decomposition process of an ultrasonic signal. S is the original signal, and a10-a37 are the wavelet coefficients.

**Figure 3 sensors-24-01440-f003:**
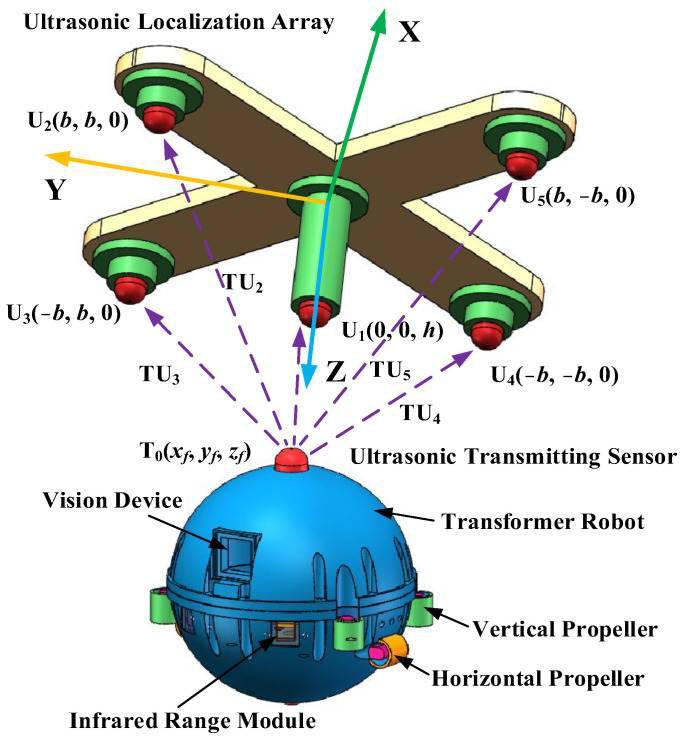
Schematic diagram of five-element ultrasound array localization.

**Figure 4 sensors-24-01440-f004:**
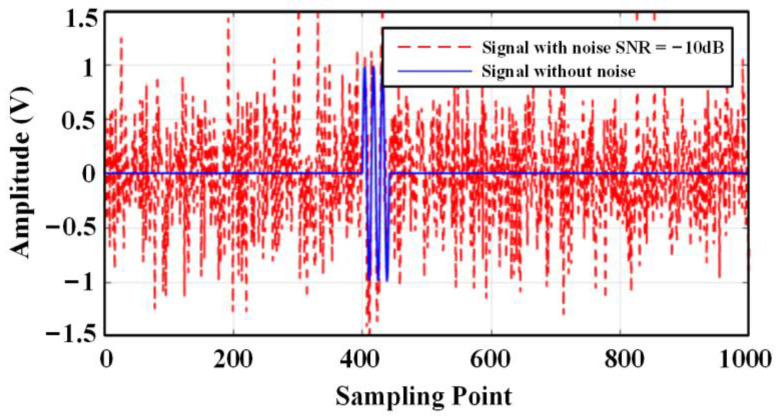
Simulated signal with a signal-to-noise ratio of −10 dB.

**Figure 5 sensors-24-01440-f005:**
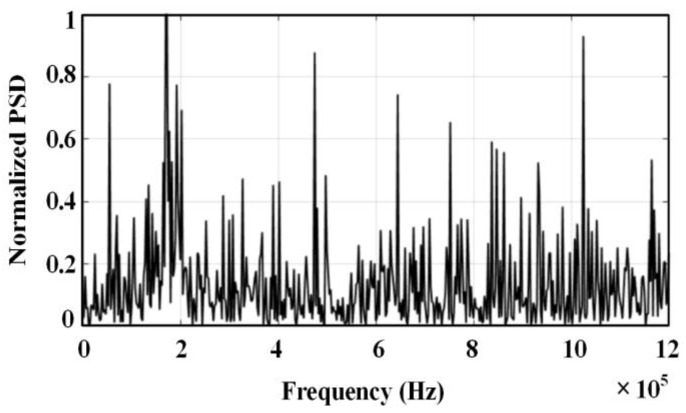
Spectrogram of the simulated ultrasonic signal.

**Figure 6 sensors-24-01440-f006:**
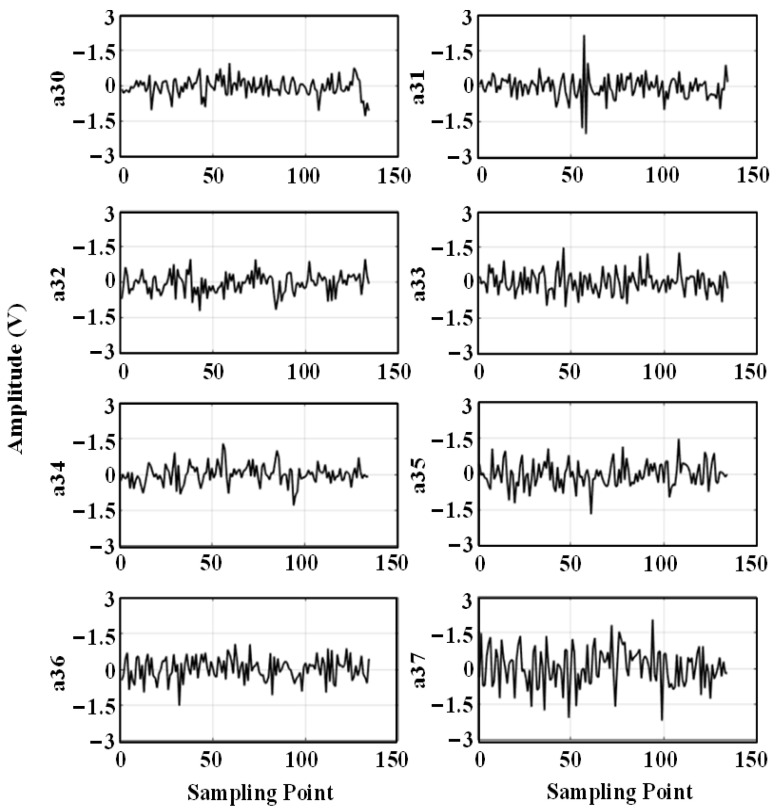
Wavelet packet decomposition feature map of simulated signal 1.

**Figure 7 sensors-24-01440-f007:**
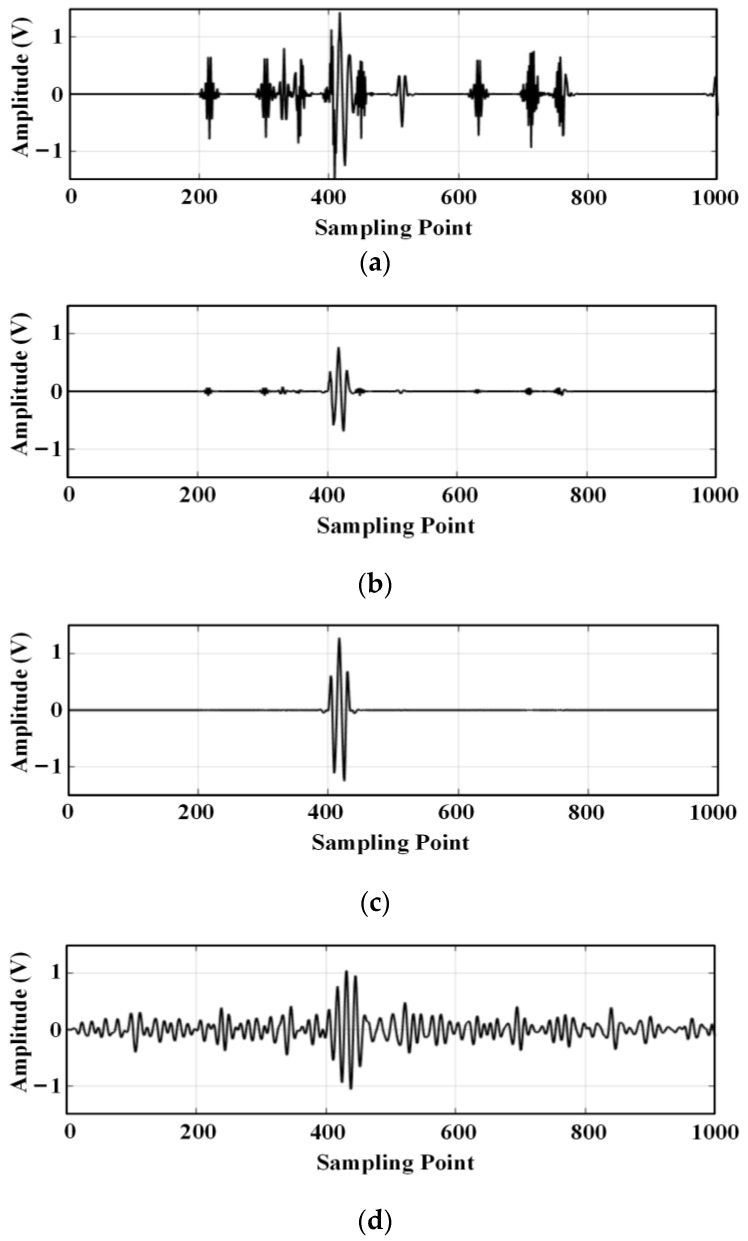
Comparison of the signal filtering effect: (**a**) Signal denoising and reconstruction with a hard threshold function; (**b**) Signal denoising and reconstruction with a soft threshold function; (**c**) Signal denoising and reconstruction with a semi-soft threshold function; (**d**) Filtering effect with a band-pass filter.

**Figure 8 sensors-24-01440-f008:**
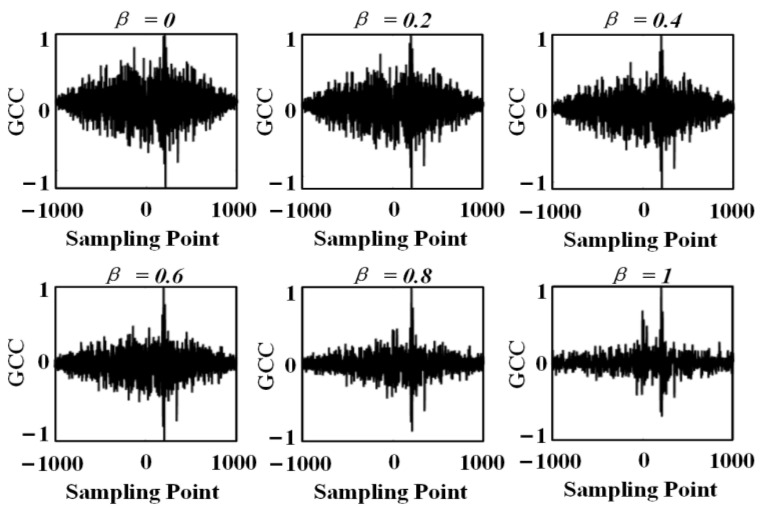
Time delay estimation of PHAT-β generalized correlation with different β values.

**Figure 9 sensors-24-01440-f009:**
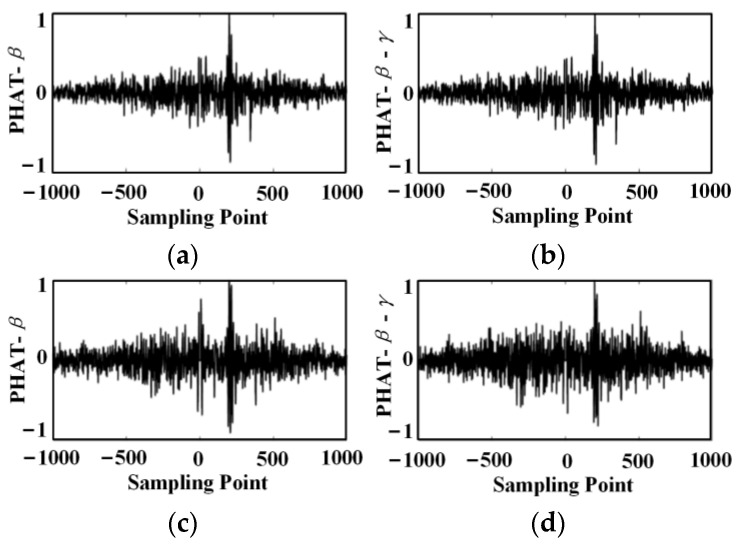
Comparison of time delay estimates for PHAT-β and PHAT-β-γ generalized cross correlation, β = 0.8 and (**a**) PHAT-β (signal amplitude 1 V); (**b**) PHAT-β-γ (signal amplitude 1 V); (**c**) PHAT-β (signal amplitude 0.1 V); (**d**) PHAT-β-γ (signal amplitude 0.1 V).

**Figure 10 sensors-24-01440-f010:**
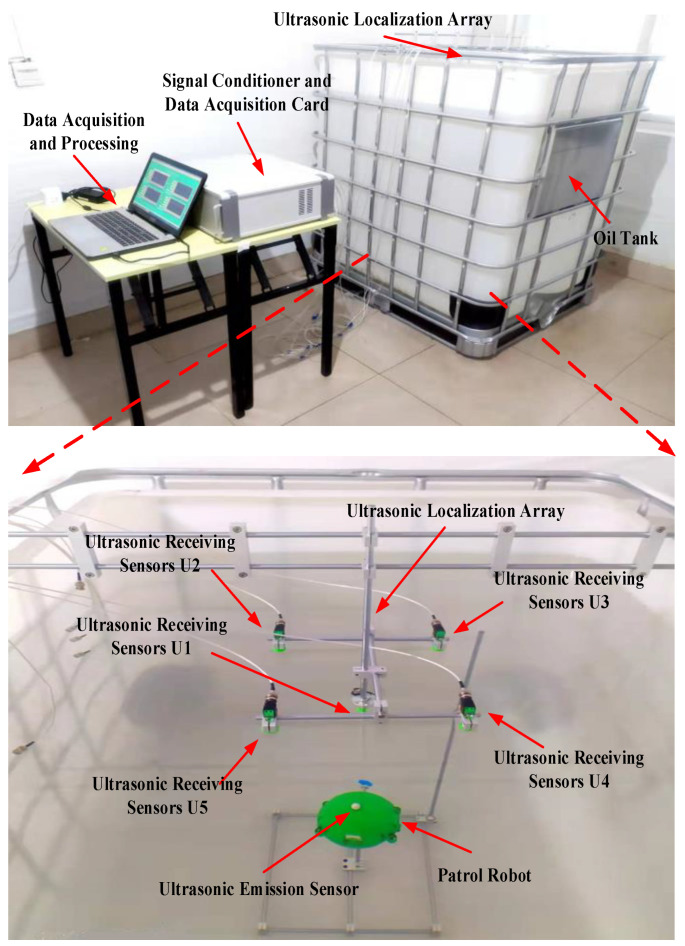
Test platform for patrol micro-robot three-dimensional spatial localization.

**Figure 11 sensors-24-01440-f011:**
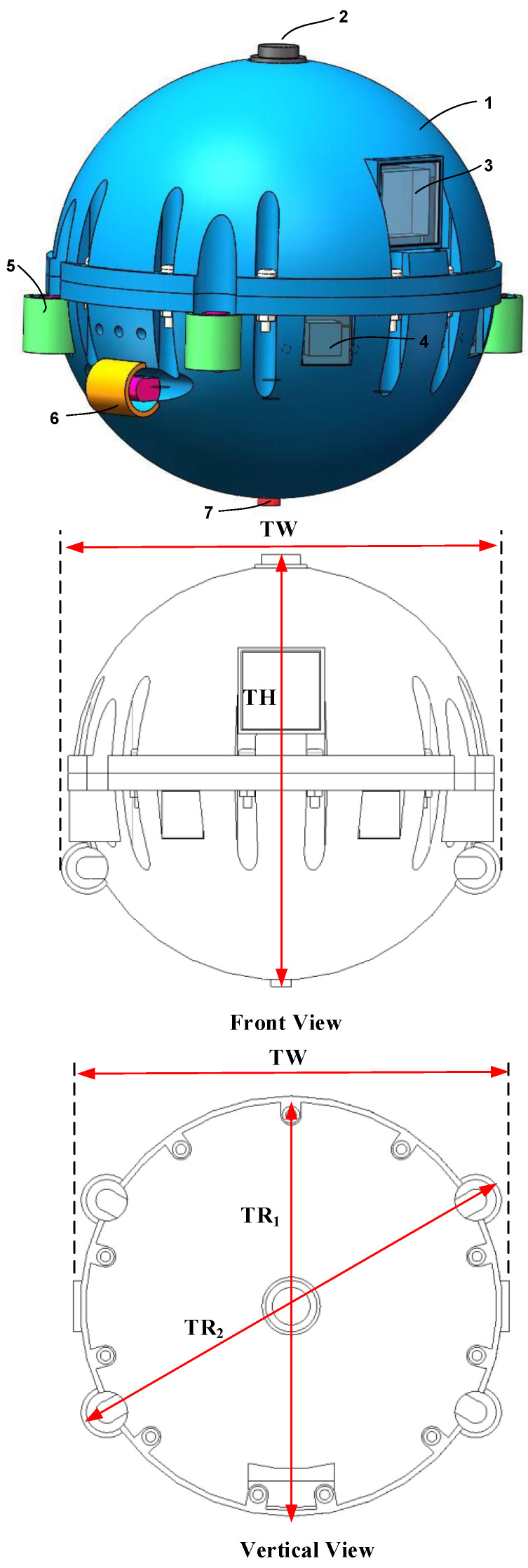
Overall structure of the transformer patrol micro-robot. #1—robot main body shell, #2—ultrasonic emission sensor, #3—vision device, #4—infrared range module, #5—robot vertical propeller propulsion device, #6—robot horizontal propeller propulsion device, #7—pressure sensor, and robot control system.

**Figure 12 sensors-24-01440-f012:**
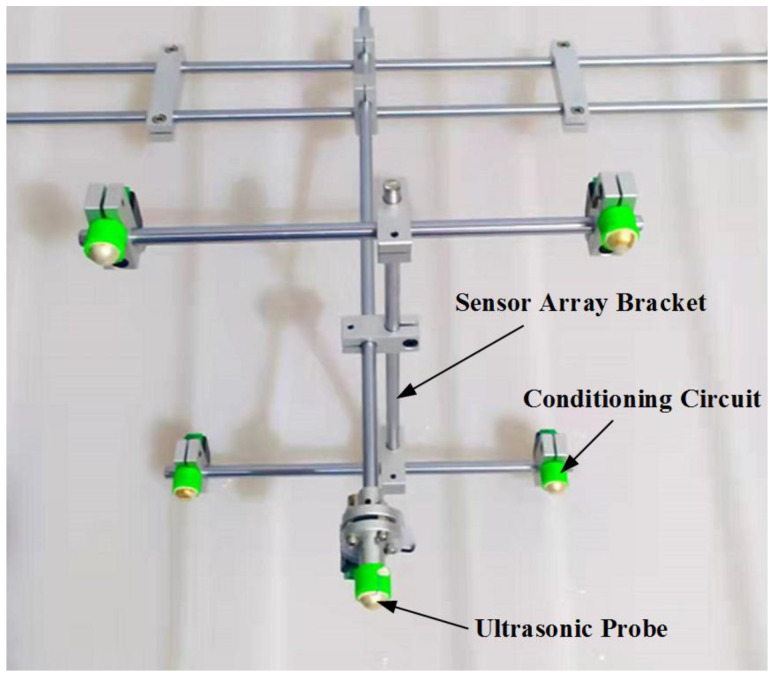
Physical diagram of an ultrasonic localization transducer array.

**Figure 13 sensors-24-01440-f013:**
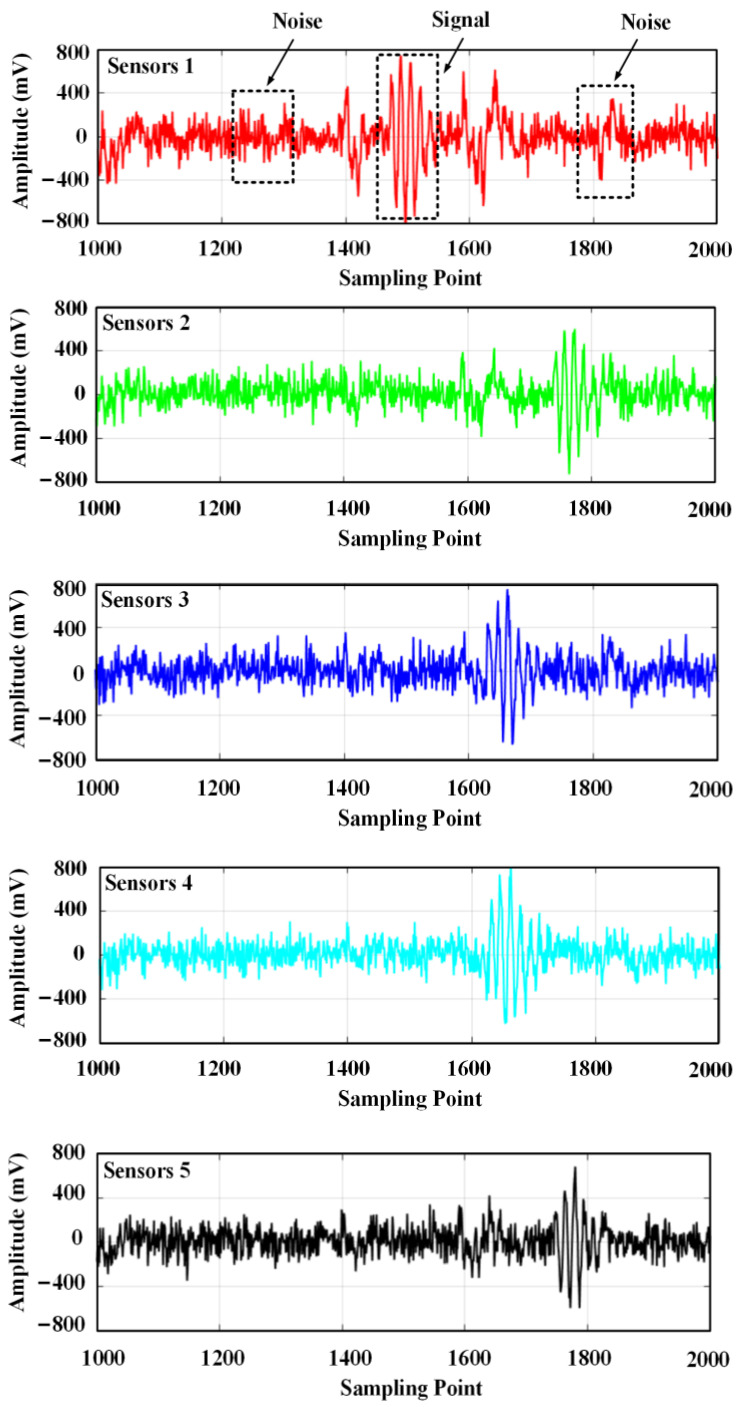
Ultrasonic signal waveform collected by the transducer unit.

**Figure 14 sensors-24-01440-f014:**
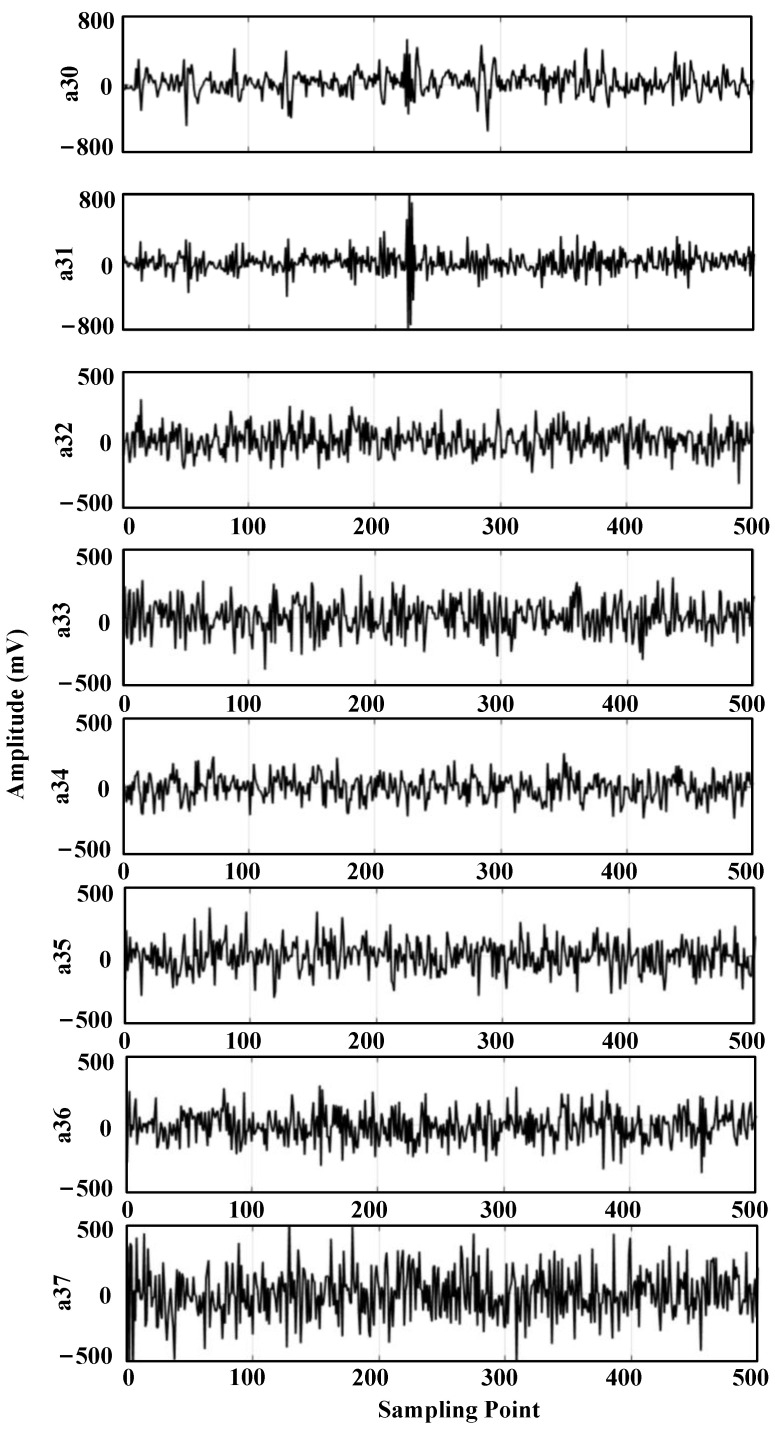
Wavelet packet decomposition coefficients of ultrasonic signals.

**Figure 15 sensors-24-01440-f015:**
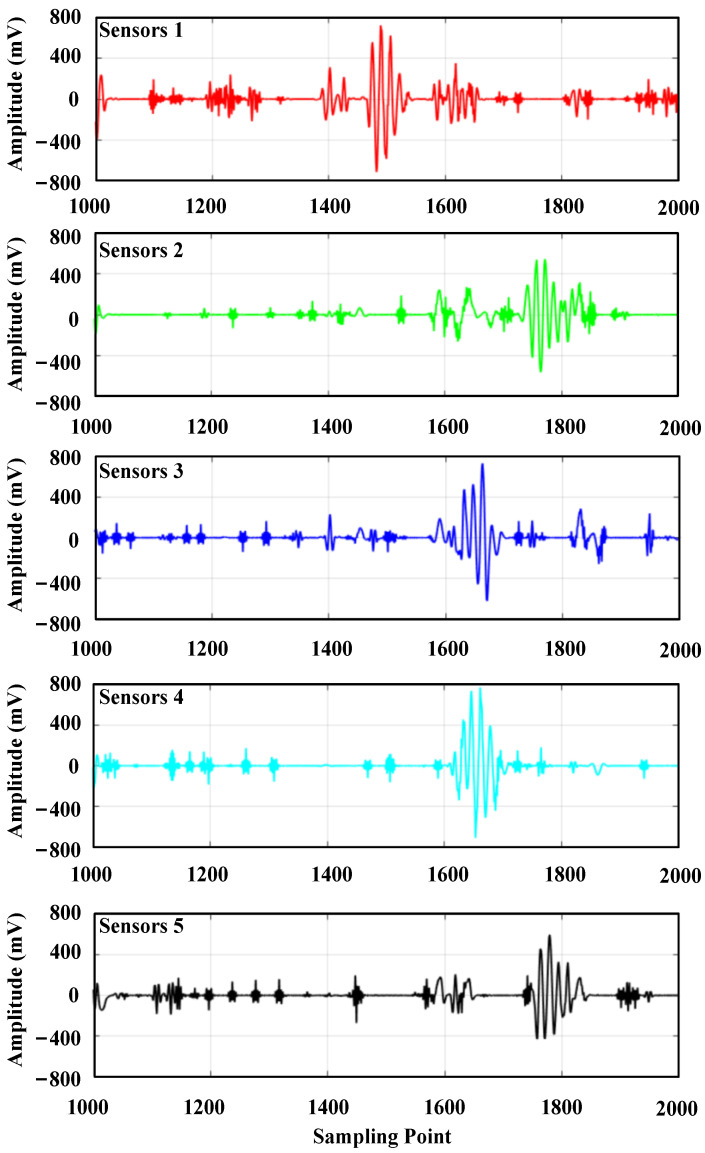
Reconstructed waveform with wavelet packet denoising.

**Figure 16 sensors-24-01440-f016:**
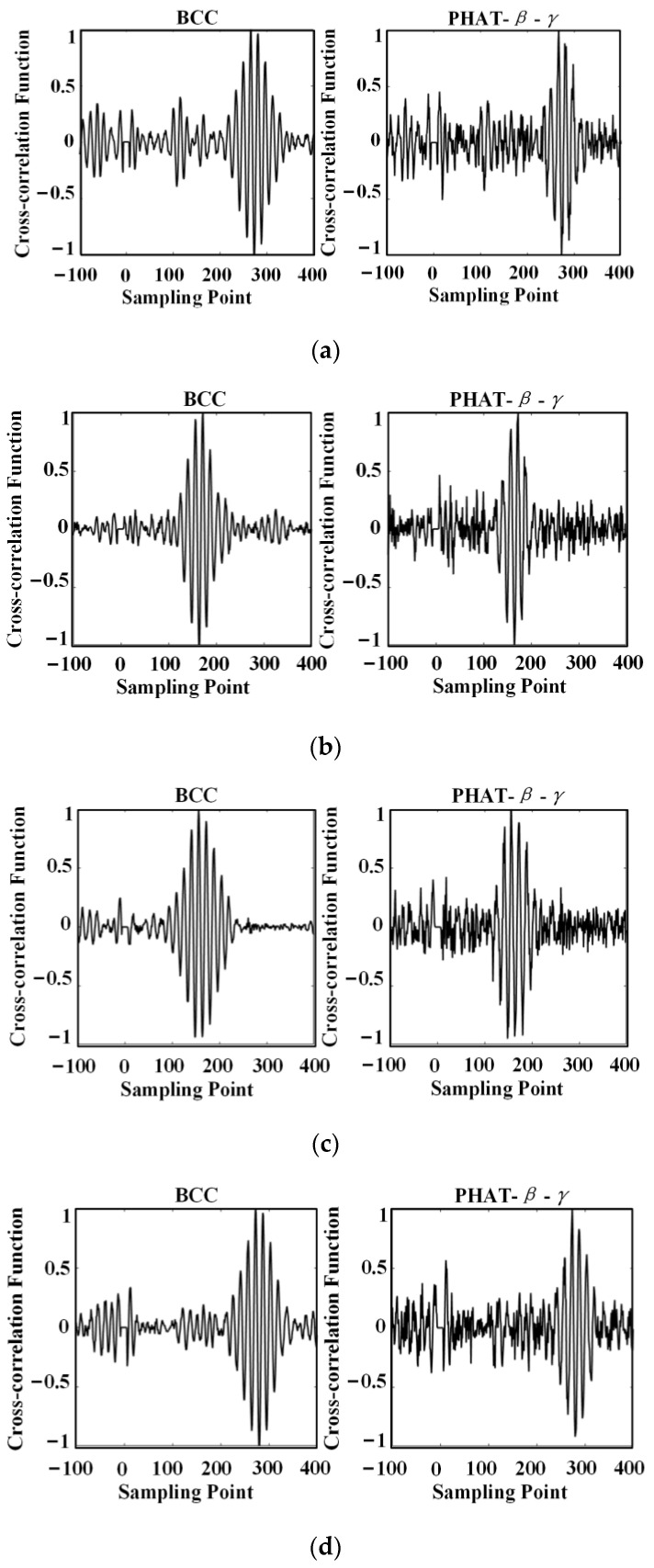
Comparison of time delay estimation results and (**a**) Sensor No. 2 and No. 1; (**b**) Sensor No. 3 and No. 1; (**c**) Sensor No. 4 and No. 1; (**d**) Sensor No. 5 and No. 1.

**Table 1 sensors-24-01440-t001:** Comparison table of signal denoising effect evaluation.

Threshold Function	SNR/dB	RMSE	NCC
Raw noisy signal	−10	0.456	0.298
Hard threshold function	−1.484	0.171	0.584
Soft threshold function	3.845	0.093	0.830
Semi-soft threshold function	5.016	0.081	0.831
Band-pass filtering	−1.259	0.167	0.564
Band-pass filtering	SNR/dB	RMSE	NCC

**Table 2 sensors-24-01440-t002:** Localization Results for Position 1.

Δt21 (us)	Δt31 (us)	Δt41 (us)	Δt51 (us)	Results (mm)	Absolute Error (mm)	Relative Localization Error
106.8	63.2	62.8	109.6	(−215, −1, 680)	(5, 1, 10)	1.55%
106.6	63.4	63.4	110.0	(−212, 0, 678)	(8, 0, 12)	1.99%
106.2	63.2	62.8	109.2	(−219, −1, 700)	(1, 0, 10)	1.39%
107.6	64.0	62.4	109.6	(−216, −7, 691)	(4, 7, 1)	1.12%
107.2	64.0	63.2	109.6	(−208, −3, 663)	(12, 3, 27)	4.10%
106.8	63.2	62.4	109.6	(−216, 0, 706)	(4, 0, 16)	2.28%
107.2	63.6	62.4	109.2	(−215, −5, 679)	(5, 5, 11)	1.81%
108.2	63.5	63.2	109.6	(−212, −1, 676)	(8, 1, 14)	2.23%
106.8	63.6	63.2	108.8	(−218, −1, 712)	(2, 1, 22)	3.05%
108.4	63.4	63.4	109.2	(−219, 0, 714)	(1, 0, 24)	3.32%

**Table 3 sensors-24-01440-t003:** Localization Results for Position 2.

Δt21 (us)	Δt31 (us)	Δt41 (us)	Δt51 (us)	Results (mm)	Absolute Error (mm)	Relative Localization Error
90.8	91.2	90.0	92.8	(−12, −5, 668)	(12, 5, 22)	3.53%
91.2	91.2	89.2	92.0	(−11, −9, 699)	(11, 9, 9)	2.44%
91.2	90.8	89.6	92.4	(−12, −5, 690)	(12, 5, 0)	1.88%
90.8	91.2	89.6	92.4	(−11, −7, 686)	(11, 7, 4)	1.98%
91.2	90.8	89.2	92.0	(−12, −7, 704)	(12, 7, 14)	2.86%
90.8	91.6	89.2	92.4	(−5, −13, 708)	(5, 13, 18)	3.30%
91.2	91.2	89.2	91.6	(−10, −9, 706)	(10, 9, 16)	3.03%
90.4	91.2	89.2	92.0	(−11, −9, 699)	(11, 9, 9)	2.44%
90.8	90.2	89.6	92.4	(−12, −2, 707)	(12, 2, 17)	3.48%
90.8	90.8	89.2	92.4	(−14, −7, 690)	(14, 7, 0)	2.27%

**Table 4 sensors-24-01440-t004:** Localization Results for Position 3.

Δt21 (us)	Δt31 (us)	Δt41 (us)	Δt51 (us)	Results (mm)	Absolute Error (mm)	Relative Localization Error
63.4	109.2	109.4	63.2	(214, 0, 688)	(6, 0, 2)	0.87%
63.4	108.8	109.0	63.2	(218, 0, 708)	(2, 0, 18)	2.50%
62.8	109.2	109.4	62.8	(222, 0, 707)	(2, 0, 17)	2.36%
63.0	109.2	109.0	62.8	(222, −1, 711)	(2, 1, 21)	2.91%
63.0	109.2	109.0	62.4	(223, −2, 711)	(3, 2, 21)	2.94%
63.8	109.6	109.0	63.2	(214, −3, 691)	(6, 3, 1)	0.94%
63.4	109.2	108.6	62.8	(222, −3, 718)	(2, 3, 28)	3.90%
63.4	109.2	109.8	62.8	(211, −2, 663)	(9, 2, 27)	3.94%
63.0	109.2	109.4	62.8	(219, 0, 697)	(1, 0, 7)	0.98%
63.4	108.8	109.0	63.2	(218, 0, 708)	(2, 0, 18)	2.5%

## Data Availability

The data presented in this study are available on request from the corresponding author. The data are not publicly available due to privacy.

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
