# Peer review of "Spatial Localization of a Transformer Robot Based on Ultrasonic Signal Wavelet Decomposition and PHAT-β-γ Generalized Cross Correlation"

_sensors, 2024, doi:10.3390/s24051440_

Round 1

Reviewer 1 Report

Comments and Suggestions for Authors

The manuscript is written on a relevant and interesting topic and is definitely of interest to the scientific community. But in my opinion, it would be possible to improve the manuscript by expanding the overview of methods and technical means used to diagnose the operation of power equipment. I also recommend adding links to similar works. To pay attention to the novelty of the research presented in the work and reflect how the results obtained by the authors correlate with similar studies. In the experimental part, a plastic tank was used, but many modern transformers are in metal boxes, I think it is necessary to reflect in the manuscript how their robot will work in other conditions and how can the influence of metal boxes on the results of the robot's work be assessed? It would be nice to provide a more detailed justification for the choice of ultrasonic sensors used by the authors and justify their characteristics.

Reviewer 2 Report

Comments and Suggestions for Authors

To achieve accurate location of patrol micro-robot inside the transformer, a spatial ultrasonic localization method based on wavelet decomposition and 14 PHAT-β-γ generalized cross correlation is proposed in this paper. However, the paper has the following problems:

1. The paper is missing axis titles in the figures 6 and 13.

2. There are too many formulas in the paper, so it is suggested to simplify them according to their importance.

3. There is a lack of description of denoising methods, and it is suggested to add the necessity description of denoising and reconstruction, and compare the signal changes before and after denoising.

4. The types of transformer oil are not single, and there is no comparative test in different types of transformer oil, so it is impossible to judge the influence effect of ultrasonic attenuation change in different media on this method.

5. Robot positioning needs real-time, and lacks positioning tests in the process of robot movement.

6. The internal environment of transformer is complex, but the oil tank used in the test is relatively regular. How does the author consider it?

7. Whether the internal structure of the transformer cause too many echoes, and whether the echoes be effectively processed by the current denoising methods?

Round 2

Reviewer 2 Report

Comments and Suggestions for Authors

The author has gived a detailed reply to the comments and suggestions.